# ON FORMAL FEATURE ATTRIBUTION AND ITS APPROXIMATION

## ABSTRACT

Recent years have witnessed the widespread use of artificial intelligence (AI) algorithms and machine learning (ML) models. Despite their tremendous success, a number of vital problems like ML model brittleness, their fairness, and the lack of interpretability warrant the need for the active developments in explainable artificial intelligence (XAI) and formal ML model verification. The two major lines of work in XAI include *feature selection* methods, e.g. Anchors, and *feature attribution* techniques, e.g. LIME and SHAP. Despite their promise, most of the existing feature selection and attribution approaches are susceptible to a range of critical issues, including explanation unsoundness and *out-of-distribution* sampling. A recent formal approach to XAI (FXAI) although serving as an alternative to the above and free of these issues suffers from a few other limitations. For instance and besides the scalability limitation, the formal approach is unable to tackle the feature attribution problem. Additionally, a formal explanation despite being formally sound is typically quite large, which hampers its applicability in practical settings. Motivated by the above, this paper proposes a way to apply the apparatus of formal XAI to the case of feature attribution based on formal explanation enumeration. Formal feature attribution (FFA) is argued to be advantageous over the existing methods, both formal and non-formal. Given the practical complexity of the problem, the paper then proposes an efficient technique for approximating exact FFA. Finally, it offers experimental evidence of the effectiveness of the proposed approximate FFA in comparison to the existing feature attribution algorithms not only in terms of feature importance and but also in terms of their relative order.[1]

## 1 INTRODUCTION

Thanks to the unprecedented fast growth and the tremendous success, Artificial Intelligence (AI) and Machine Learning (ML) have become a universally acclaimed standard in automated decision making causing a major disruption in computing and the use of technology in general (LeCun et al., 2015; Jordan and Mitchell, 2015; Mnih et al., 2015; ACM, 2018). An ever growing range of practical applications of AI and ML, on the one hand, and a number of critical issues observed in modern AI systems (e.g. decision bias (Angwin et al., 2016) and brittleness (Szegedy et al., 2014)), on the other hand, gave rise to the quickly advancing area of theory and practice of Explainable AI (XAI).

Numerous methods exist to explain decisions made by what is called black-box ML models (Miller, 2019; Molnar, 2020). Here, *model-agnostic* approaches based on random sampling prevail (Miller, 2019), with the most popular being *feature selection* (Ribeiro et al., 2018) and *feature attribution* (Lundberg and Lee, 2017; Ribeiro et al., 2018) approaches. Despite their promise, model-agnostic approaches are susceptible to a range of critical issues, like unsoundness of explanations (Ignatiev, 2020; Huang and Marques-Silva, 2023) and *out-of-distribution sampling* (Slack et al., 2020; Lakkaraju and Bastani, 2020), which exacerbates the problem of trust in AI.

An alternative to model-agnostic explainers is represented by the methods building on the success of formal reasoning applied to the logical representations of ML models (Shih et al., 2018; Marques-Silva and Ignatiev, 2022). Aiming to address the limitations of model-agnostic approaches, formal XAI (FXAI) methods themselves suffer from a few downsides, including the lack of scalability and

---

[1]Source code and complete experimental setup are available in the supplementary material.

Figure 1: Example BT model (Chen and Guestrin, 2016) trained on the *adult* classification dataset.

the requirement to build a complete logical representation of the ML model. Formal explanations also tend to be larger than their model-agnostic counterparts because they do not reason about (unknown) data distribution (Wäldchen et al., 2021). Finally and most importantly, FXAI methods have not been applied so far to answer feature attribution questions.

Motivated by the above, we define a novel formal approach to feature attribution, which builds on the success of existing FXAI methods (Marques-Silva and Ignatiev, 2022). By exhaustively enumerating all formal explanations, we can give a crisp definition of *formal feature attribution* (FFA) as the proportion of explanations in which a given feature occurs. We argue that formal feature attribution is hard for the second level of the polynomial hierarchy. Although it can be challenging to compute exact FFA in practice, we show that existing anytime formal explanation enumeration methods can be applied to efficiently approximate FFA. Our experimental results demonstrate the effectiveness of the proposed approach in practice and its advantage over LIME and a few variants of SHAP given publicly available tabular and image datasets, as well as on a real application of XAI in the domain of Software Engineering (McIntosh and Kamei, 2017; Pornprasit et al., 2021).

## 2 BACKGROUND

This section briefly overviews the status quo in XAI and background knowledge the paper builds on.

### 2.1 CLASSIFICATION PROBLEMS

Classification problems consider a set of classes $\mathcal{K} = \{1, 2, \ldots, k\}^2$, and a set of features $\mathcal{F} = \{1, \ldots, m\}$. The value of each feature $i \in \mathcal{F}$ is taken from a domain $\mathbb{D}_i$, which can be categorical or ordinal, i.e. integer, real-valued or Boolean. Therefore, the complete feature space is defined as $\mathbb{F} \triangleq \prod_{i=1}^{m} \mathbb{D}_i$. A concrete point in feature space is represented by $\mathbf{v} = (v_1, \ldots, v_m) \in \mathbb{F}$, where each component $v_i \in \mathbb{D}_i$ is a constant taken by feature $i \in \mathcal{F}$. An *instance* or *example* is denoted by a specific point $\mathbf{v} \in \mathbb{F}$ in feature space and its corresponding class $c \in \mathcal{K}$, i.e. a pair $(\mathbf{v}, c)$ represents an instance. Additionally, the notation $\mathbf{x} = (x_1, \ldots, x_m)$ denotes an arbitrary point in feature space, where each component $x_i$ is a variable taking values from its corresponding domain $\mathbb{D}_i$ and representing feature $i \in \mathcal{F}$. A classifier defines a non-constant classification function $\kappa : \mathbb{F} \to \mathcal{K}$.

Many ways exist to learn classifiers $\kappa$ given training data, i.e. a collection of labeled instances $(\mathbf{v}, c)$, including decision trees (Hyafil and Rivest, 1976) and their ensembles (Breiman, 2001; Chen and Guestrin, 2016), decision lists (Rivest, 1987), neural networks (LeCun et al., 2015), etc. This paper considers boosted tree (BT) models trained with the use of XGBoost (Chen and Guestrin, 2016).

**Example 1.** *Figure 1 shows a BT model trained for a simplified version of the* adult *dataset (Kohavi, 1996). For an instance* $\mathbf{v} = \{Education = Bachelors, Status = Separated, Occupation = Sales, Relationship = Not-in-family, Sex = Male, Hours/w \leq 40\}$, *the model predicts* <50k *because the sum of the weights in the 3 trees for this instance equals* $-0.4073 = (-0.1089 - 0.2404 - 0.0580) < 0$.

### 2.2 ML MODEL INTERPRETABILITY AND POST-HOC EXPLANATIONS

Interpretability is usually deemed to be a subjective concept, with no formal definition (Lipton, 2018). One way to measure interpretability is in terms of the succinctness of information provided by an ML

---

[2]Any set of classes $\{c_1, \ldots, c_k\}$ can always be mapped into the set of the corresponding indices $\{1, \ldots, k\}$.

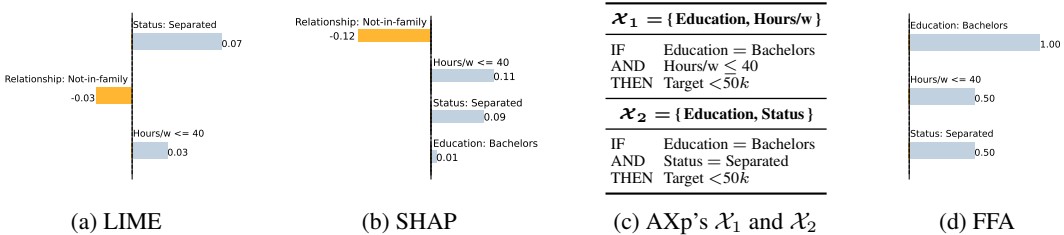

Figure 2: Examples of feature attribution reported by LIME and SHAP, as well as both AXp's (no more AXp's exist) followed by FFA for the instance **v** shown in Example 1.

model to justify a given prediction. Recent years have witnessed an upsurge in the interest in applying interpretable models in safety-critical applications (Rudin, 2019; Molnar, 2020). An alternative to interpretable models is post-hoc explanation of *black-box* models, which this paper focuses on.

Numerous methods to compute explanations have been proposed recently (Miller, 2019; Molnar, 2020). The lion's share of these comprise what is called *model-agnostic* approaches to explainability (Ribeiro et al., 2016; Lundberg and Lee, 2017; Ribeiro et al., 2018) are of heuristic nature that resort to extensive sampling in the vicinity of an instance to explain in order to "estimate" the behavior of the classifier in this local vicinity of the instance. In this regard, they rely on estimating input data distribution by building on the information about the training data (Lakkaraju and Bastani, 2020). Depending on the form of explanations model-agnostic approaches offer, they are conventionally classified as *feature selection* or *feature attribution* approaches briefly discussed below.

**Feature Selection** approaches identify feature subsets that are deemed *sufficient* for a given prediction $c = \kappa(\mathbf{v})$. As such, a feature selection explanation given as a set of features $\omega \subseteq \mathcal{F}$ should be interpreted as the conjunction $\bigwedge_{i \in \omega} (x_i = v_i)$ deemed responsible for prediction $c = \kappa(\mathbf{v})$, $\mathbf{v} \in \mathbb{F}$, $c \in \mathcal{K}$. The majority of feature selection approaches are model-agnostic with one prominent example being Anchors (Ribeiro et al., 2018). As such, the sufficiency of the selected set of features for a given prediction is determined statistically based on extensive sampling around the instance of interest, by assessing a few measures like *fidelity*, *precision*, among others. Due to the statistical nature of these explainers, they are known to suffer from various explanation quality issues (Lakkaraju and Bastani, 2020; Ignatiev, 2020; Slack et al., 2021). An additional line of work on *formal* explainability also tackles feature selection while offering guarantees of soundness; these are discussed below.

**Feature Attribution.** A different view on post-hoc explanations is provided by feature attribution approaches, e.g. LIME (Ribeiro et al., 2016) and SHAP (Lundberg and Lee, 2017). Based on random sampling in the neighborhood of the target instance, these approaches attribute responsibility to all model's features by assigning a numeric value $w_i \in \mathbb{R}$ of importance to each feature $i \in \mathcal{F}$. Given these importance values, the features can then be ranked from most important to least important. As a result, a feature attribution explanation is conventionally provided as a linear form $\sum_{i \in \mathcal{F}} w_i \cdot x_i$, which can be also seen as approximating the original black-box explainer $\kappa$ in the *local* neighborhood of instance $\mathbf{v} \in \mathbb{F}$. Among feature attribution approaches, SHAP (Lundberg and Lee, 2017; Arenas et al., 2021b;c) and its variants is often claimed to stand out as it aims at approximating Shapley values, a powerful concept originating from cooperative games in game theory (Shapley, 1953).

**Formal Explainability.** This work builds on formal explainability proposed in earlier work (Shih et al., 2018; Ignatiev et al., 2019; Darwiche and Hirth, 2020; Audemard et al., 2020; Marques-Silva and Ignatiev, 2022) where explanations are equated with *abductive explanations* (AXp's). Abductive explanations are *subset-minimal* sets of features *formally proved* to suffice to explain an ML prediction given a formal representation of the classifier of interest. Concretely, given an instance $\mathbf{v} \in \mathbb{F}$ and a prediction $c = \kappa(\mathbf{v})$, an AXp is a subset-minimal set of features $\mathcal{X} \subseteq \mathcal{F}$, such that

$$\forall(\mathbf{x} \in \mathbb{F}). \bigwedge_{i \in \mathcal{X}} (x_i = v_i) \rightarrow (\kappa(\mathbf{x}) = c) \tag{1}$$

Abductive explanations are *guaranteed* to be subset-minimal sets of features proved to satisfy (1). As other feature selection explanations, they answer *why* a certain prediction was made. An alternate way to explain a model's behavior is to seek an answer *why not* another prediction was made, or, in other words, *how* to change the prediction. Explanations answering *why not* questions are referred to

as *contrastive explanations* (CXp's) (Miller, 2019; Ignatiev et al., 2020; Marques-Silva and Ignatiev, 2022). As in prior work, we define a CXp as a subset-minimal set of features that, if allowed to change their values, are *necessary* to change the prediction of the model. Formally, a CXp for prediction $c = \kappa(\mathbf{v})$ is a subset-minimal set of features $\mathcal{Y} \subseteq \mathcal{F}$, such that

$$\exists (\mathbf{x} \in \mathbb{F}). \bigwedge_{i \notin \mathcal{Y}} (x_i = v_i) \wedge (\kappa(\mathbf{x}) \neq c) \tag{2}$$

Finally, recent work has shown that AXp's and CXp's for a given instance $\mathbf{v} \in \mathbb{F}$ enjoy the *minimal hitting set duality* (Ignatiev et al., 2020; Reiter, 1987). The duality implies that each AXp for a prediction $c = \kappa(\mathbf{v})$ is a *minimal hitting set*[3] (MHS) of the set of all CXp's for that prediction, and the other way around: each CXp is an MHS of the set of all AXp's. The explanation enumeration algorithm applied in this paper heavily relies on this duality relation and is inspired by the MARCO algorithm originating from the area of over-constrained systems (Liffiton et al., 2016). A growing body of recent work on formal explanations is represented (but not limited) by (Marques-Silva et al., 2021; Arenas et al., 2021a; Wäldchen et al., 2021; Darwiche and Marquis, 2021; Malfa et al., 2021; Boumazouza et al., 2021; Blanc et al., 2021; Gorji and Rubin, 2022; Marques-Silva and Ignatiev, 2022; Amgoud and Ben-Naim, 2022; Ferreira et al., 2022; Arenas et al., 2022).

**Example 2.** *In the context of Example 1, feature attribution computed by LIME and SHAP as well as all 2 AXp's are shown in Figure 2. AXp $\mathcal{X}_1$ indicates that specifying Education = Bachelors and Hours/w $\leq$ 40 guarantees that any compatible instance is classified as < 50k independent of the values of other features, e.g. Status and Relationship, since the maximal sum of weights is $0.0770 - 0.0200 - 0.0580 = -0.0010 < 0$ as long as the feature values above are used. Observe that another AXp $\mathcal{X}_2$ for $\mathbf{v}$ is {Education, Status}. Since both of the two AXp's for $\mathbf{v}$ consist of two features, it is difficult to judge which one is better without a formal feature importance assessment.*

## 3  WHY FORMAL FEATURE ATTRIBUTION?

On the one hand, abductive explanations serve as a viable alternative to non-formal feature selection approaches because they (i) guarantee subset-minimality of the selected sets of features and (ii) are computed via formal reasoning over the behavior of the corresponding ML model. Having said that, they suffer from a few issues. First, observe that deciding the validity of (1) requires a formal reasoner to take into account the complete feature space $\mathbb{F}$, assuming that the features are independent and uniformly distributed (Wäldchen et al., 2021). In other words, the reasoner has to check all the combinations of feature values, including those that *never appear in practice*. This makes AXp's being unnecessarily *conservative* (long), i.e. they may be hard for a human decision maker to interpret. Second, AXp's are not aimed at providing feature attribution. The abundance of various AXp's for a single data instance (Ignatiev et al., 2019), e.g. see Example 2, exacerbates this issue as it becomes unclear for a user which of the AXp's to use to make an informed decision in a particular situation.

On the other hand, non-formal feature attribution in general is known to be susceptible to out-of-distribution sampling (Lakkaraju and Bastani, 2020; Slack et al., 2020) while SHAP has been recently shown to fail to effectively approximate Shapley values (Huang and Marques-Silva, 2023). Quite surprisingly, (Huang and Marques-Silva, 2023) also argued that even the use of *exact* Shapley values may be inadequate as a measure of feature importance. Namely, they used the concept of formal *feature relevancy*, i.e. a feature is said to be *relevant* for a given prediction $c = \kappa(\mathbf{v})$ iff it appears in at least one AXp for the prediction Huang et al. (2023), and practically showed that (i) *irrelevant* features may have *non-zero* Shapley values while (ii) *relevant* features may be assigned *zero* Shapley values. Our results below confirm that both LIME and SHAP often report attributions inconsistent with relevant features for the corresponding predictions in a number of practical scenarios.

To address the above limitations, we propose the concept of *formal feature attribution* (FFA). (An insight on this was also given in (Huang and Marques-Silva, 2023).) Let us denote the set of all formal AXp's for a prediction $c = \kappa(\mathbf{v})$ by $\mathbb{A}_\kappa(\mathbf{v}, c)$. Then formal feature attribution of a feature $i \in \mathcal{F}$ can be defined as the proportion of abductive explanations where it occurs. More formally,

**Definition 1: (FFA).** The *formal feature attribution* $\mathrm{ffa}_\kappa(i, (\mathbf{v}, c))$ of a feature $i \in \mathcal{F}$ to an instance $(\mathbf{v}, c)$ for machine learning model $\kappa$ is as follows: $\mathrm{ffa}_\kappa(i, (\mathbf{v}, c)) = |\{\mathcal{X} \mid \mathcal{X} \in \mathbb{A}_\kappa(\mathbf{v},c), i \in \mathcal{X}\}| / |\mathbb{A}_\kappa(\mathbf{v},c)|$.

---

[3]Given a set of sets $\mathbb{S}$, a *hitting set* of $\mathbb{S}$ is a set $H$ such that $\forall S \in \mathbb{S}, S \cup H \neq \emptyset$, i.e. $H$ "hits" every set in $\mathbb{S}$. A hitting set $H$ for $\mathbb{S}$ is *minimal* if none of its strict subsets is also a hitting set.

Formal feature attribution has some nice properties. First, it has a strict and formal definition, i.e. we can, assuming we are able to compute the complete set of AXp's for an instance, exactly define it for all features $i \in \mathcal{F}$. Second, it is fairly easy to explain to a user of the classification system, even if they are non-expert. Indeed, it is the percentage of (formal abductive) explanations that make use of a particular feature $i$. Third, as we shall see later, even though we may not be able to compute all AXp's exhaustively, we can still get good FFA approximations fast. Fourth, since it is based on formal explanations, it is immune from out-of-distribution sampling problems.

**Example 3.** *Recall Example 2. As there are 2 AXp's for instance* **v***, the prediction can be attributed to the 3 features with non-zero FFA shown in Figure 2d. Also, observe how both LIME and SHAP (see Figure 2a and Figure 2b) assign non-zero attribution to the feature Relationship, which is in fact* irrelevant *for the prediction, but overlook the highest importance of feature Education.*

One criticism of the above definition is that it does not take into account the length of explanations where the feature arises. Arguably if a feature arises in many AXp's of size 2, it should be considered more important than a feature which arises in the same number of AXp's but where each is of size 10. An alternate definition, which tries to take this into account, is the weighted formal feature attribution (WFFA), i.e. the *average* proportion of AXp's that include feature $i \in \mathcal{F}$. Formally,

**Definition 2: (WFFA).** The *weighted formal feature attribution* $\text{wffa}_\kappa(i, (\mathbf{v}, c))$ of a feature $i \in \mathcal{F}$ to an instance $(\mathbf{v}, c)$ for machine learning model $\kappa$ is $\text{wffa}_\kappa(i, (\mathbf{v}, c)) = \sum_{\mathcal{X} \in \mathbb{A}_\kappa(\mathbf{v}, c), i \in \mathcal{X}} |\mathcal{X}|^{-1} / |\mathbb{A}_\kappa(\mathbf{v}, c)|$.

Note that these attribution values are not on the same scale although they are convertible:

$$\sum_{i \in \mathcal{F}} \text{ffa}_\kappa(i, (\mathbf{v}, c)) = \frac{\sum_{\mathcal{X} \in \mathbb{A}_\kappa(\mathbf{v}, c)} |\mathcal{X}|}{|\mathbb{A}_\kappa(\mathbf{v}, c)|} \times \sum_{i \in \mathcal{F}} \text{wffa}_\kappa(i, (\mathbf{v}, c)).$$

Although WFFA in theory better reflects feature importance than FFA, our practical results suggest that the size of AXp's is tightly clustered around the mean, which makes the values of FFA and WFFA almost indistinguishable. For this reason and due to simplicity of the unweighted variant from a user's perspective, our experimental results focus solely on FFA (the appendix details WFFA results).

Importantly, both FFA and WFFA, by definition, respect feature relevancy (Huang et al., 2023) as feature $i \in \mathcal{F}$ is relevant for prediction $c = \kappa(\mathbf{v})$ if and only if $\text{ffa}_\kappa(i, (\mathbf{v}, c)) > 0$. Furthermore,

**Proposition 1.** *Given a feature $i \in \mathcal{F}$ and a prediction $c = \kappa(\mathbf{v})$, deciding whether $\text{ffa}_\kappa(i, (\mathbf{v}, c)) > \omega$, $\omega \in (0, 1]$, is at least as hard as deciding whether feature $i$ is relevant for the prediction.* □

This means that computing exact FFA values may be expensive in practice. For example and in light of (Huang et al., 2023), the decision version of the problem is $\Sigma_2^P$-hard in the case of DNF classifiers.

Using the relation between FFA and feature relevancy above, we can also note that the decision version of the problem is in $\Sigma_2^P$ as long as deciding the validity of (1) is in NP, which in general is the case. (Deciding (1) may be simpler, e.g. for decision trees Izza et al. (2022).) The following result is a simple consequence of the membership result for the feature relevancy problem Huang et al. (2023).

**Proposition 2.** *Deciding whether $\text{ffa}_\kappa(i, (\mathbf{v}, c)) > 0$ is in $\Sigma_2^P$ if deciding (1) is in NP.* □

## 4 Approximating Formal Feature Attribution

It may be challenging in practice to compute exact FFA due to the general complexity of the problem. Although some ML models admit efficient formal encodings and reasoning procedures, effective principal methods for FFA approximation seem necessary. This section proposes one such method.

Normally, formal explanation enumeration is done by exploiting the MHS duality between AXp's and CXp's and the use of MARCO-like (Liffiton et al., 2016) algorithms aiming at efficient exploration of minimal hitting sets of either AXp's or CXp's (Liffiton and Malik, 2013; Previti and Marques-Silva, 2013; Liffiton et al., 2016; Ignatiev et al., 2020). Depending on the target type of formal explanation, MARCO exhaustively enumerates all such explanations one by one, each time extracting a candidate minimal hitting set and checking if it is a desired explanation. If it is then it is recorded and blocked such that this candidate is never repeated again. Otherwise, a dual explanation is extracted from the subset of features complementary to the candidate (Ignatiev et al., 2019), gets recorded and blocked

---

**Algorithm 1** MARCO-like Anytime Explanation Enumeration

1: **procedure** XPENUM($\kappa$, $\mathbf{v}$, $c$)
2: $\quad$ $(\mathbb{A}, \mathbb{C}) \leftarrow (\emptyset, \emptyset)$ $\qquad\qquad\qquad\qquad\qquad\qquad$ ▷ *Sets of AXp's and CXp's to collect.*
3: $\quad$ **while** true **do**
4: $\quad\quad$ $\mathcal{Y} \leftarrow$ MINIMALHS($\mathbb{A}$, $\mathbb{C}$) $\qquad\qquad\qquad\qquad$ ▷ *Get a new MHS of $\mathbb{A}$ subject to $\mathbb{C}$.*
5: $\quad\quad$ **if** $\mathcal{Y} = \bot$ **then break** $\qquad\qquad\qquad\qquad\qquad$ ▷ *Stop if none is computed.*
6: $\quad\quad$ **if** $\exists(\mathbf{x} \in \mathbb{F}). \bigwedge_{i \notin \mathcal{Y}}(x_i = v_i) \wedge (\kappa(\mathbf{x}) \neq c)$ **then** $\quad$ ▷ *Check CXp condition (2) for $\mathcal{Y}$.*
7: $\quad\quad\quad$ $\mathbb{C} \leftarrow \mathbb{C} \cup \{\mathcal{Y}\}$ $\qquad\qquad\qquad\qquad\qquad$ ▷ *$\mathcal{Y}$ appears to be a CXp.*
8: $\quad\quad$ **else** $\qquad\qquad\qquad\qquad$ ▷ *There must be a missing AXp $\mathcal{X} \subseteq \mathcal{F} \setminus \mathcal{Y}$.*
9: $\quad\quad\quad$ $\mathcal{X} \leftarrow$ EXTRACTAXP($\mathcal{F} \setminus \mathcal{Y}$, $\kappa$, $\mathbf{v}$, $c$) $\quad$ ▷ *Get AXp $\mathcal{X}$ by iteratively checking (1).*
10: $\quad\quad\quad$ $\mathbb{A} \leftarrow \mathbb{A} \cup \{\mathcal{X}\}$ $\qquad\qquad\qquad\qquad\qquad$ ▷ *Collect new AXp $\mathcal{X}$.*
$\quad$ **return** $\mathbb{A}$, $\mathbb{C}$

---

so that it is hit by each future candidate. The procedure proceeds until no more hitting sets of the set of dual explanations can be extracted, which signifies that all target explanations are enumerated. While doing so, MARCO also enumerates all the dual explanations as a kind of "side effect".

One of the properties of MARCO used in our approximation approach is that it is an *anytime* algorithm, i.e. we can run it for as long as we need to get a sufficient number of explanations. This means we can stop it by using a timeout or upon collecting a certain number of explanations.

The main insight of FFA approximation is as follows. Recall that to compute FFA, we are interested in AXp enumeration. Although intuitively this suggests the use of MARCO targeting AXp's, for the sake of fast and high-quality FFA approximation, we propose to target CXp enumeration with AXp's as dual explanations computed "unintentionally". The reason for this is twofold: (i) we need to get a good FFA approximation as fast as we can and (ii) according to our practical observations, MARCO needs to amass a large number of dual explanations before it can start producing target explanations. This is because the hitting set enumerator is initially "blind" and knows nothing about the features it should pay attention to — it uncovers this information gradually by collecting dual explanations to hit. This way a large number of dual explanations can quickly be enumerated during this initial phase of grasping the search space, essentially "for free". Our experimental results demonstrate the effectiveness of this strategy in terms of monotone convergence of approximate FFA to the exact FFA with the increase of the time limit. A high-level view of the version of MARCO used in our approach targeting CXp enumeration and amassing AXp's as dual explanations is shown in Algorithm 1.

## 5 EXPERIMENTAL EVIDENCE

Here, we assess FFA for gradient boosted trees (Chen and Guestrin, 2016) on multiple widely used images and tabular datasets and compare it with LIME (Ribeiro et al., 2016), TreeSHAP (Lundberg et al., 2020), and KernelSHAP (Lundberg and Lee, 2017).[4] (The appendix also showcases the use of FFA in a real-world scenario of Just-in-Time (JIT) defect prediction (Pornprasit et al., 2021).)

**Setup and Prototype Implementation.** All experiments were performed on an Intel Xeon 8260 CPU running Ubuntu 20.04.2 LTS. An FFA computation prototype implementing Algorithm 1 was developed in Python and builds on (Ignatiev et al., 2022). As the FFA and WFFA values turn out to be almost identical (subject to normalization) in our experiments, here we report only (unweighted) FFA.

**Datasets and Machine Learning Models.** The well-known MNIST dataset (Deng, 2012; Paszke et al., 2019) of hand-written digits 0–9 is considered, with two concrete binary classification tasks created: 1 vs. 3 and 1 vs. 7. We also consider PneumoniaMNIST (Yang et al., 2023), a binary classification dataset to distinguish X-ray images of pneumonia from normal cases. To demonstrate extraction of *exact* FFA values for the above datasets, we also examine their downscaled versions, i.e. reduced from $28 \times 28 \times 1$ to $10 \times 10 \times 1$. Furthermore, we examine the CIFAR-10 image dataset, and the detailed results for this dataset can be found in the appendix. We also consider 11 tabular datasets often applied in the area of ML explainability and fairness (Olson et al., 2017; Dua and Graff, 2017; Schmidt and Witte, 1988; Angwin et al., 2016; FairML; Friedler et al., 2015). All the

---

[4]LIME, TreeSHAP, and KernelSHAP are sampling-based feature attribution methods. LIME and KernelSHAP are model-agnostic while TreeSHAP is specifically designed to effectively address tree-based models.

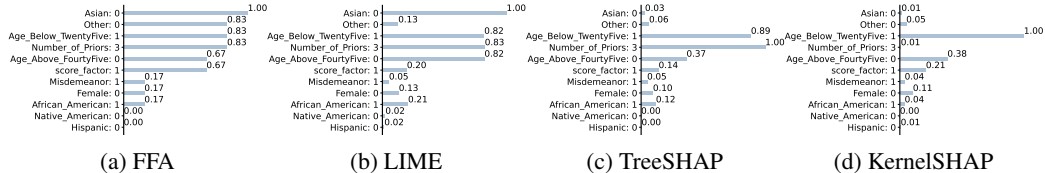

|     (a) FFA     |     (b) LIME     |     (c) TreeSHAP     |     (d) KernelSHAP     |

Figure 3: Explanations for a randomly selected instance of the Compas dataset.

Table 1: LIME, TreeSHAP, and KernalSHAP versus FFA on tabular data.

| Dataset ($|\mathcal{F}|$) | adult (12) | appendicitis (7) | australian (14) | cars (8) | compas (11) | heart-statlog (13) | hungarian (13) | lending (9) | liver-disorder (6) | pima (8) | recidivism (15) |
|---|---|---|---|---|---|---|---|---|---|---|---|
| **Approach** | | | | | | **Error** | | | | | |
| LIME | 4.48 | 2.25 | 5.13 | 1.53 | 3.28 | 4.48 | 4.56 | 1.39 | 2.39 | 2.72 | 4.73 |
| TreeSHAP | 4.47 | 2.01 | 4.49 | 1.40 | 2.67 | 3.71 | 4.14 | 1.44 | 2.28 | 3.00 | 4.76 |
| KernelSHAP | 4.32 | 2.13 | 4.60 | 0.83 | 2.59 | 3.55 | 4.02 | 1.34 | 2.23 | 2.95 | 4.81 |
| | | | | | | **Kendall's Tau** | | | | | |
| LIME | 0.07 | 0.11 | 0.22 | -0.11 | -0.11 | 0.17 | 0.04 | -0.36 | -0.22 | 0.17 | 0.05 |
| TreeSHAP | 0.03 | 0.12 | 0.27 | -0.10 | -0.10 | 0.17 | 0.20 | -0.39 | -0.21 | 0.07 | 0.12 |
| KernelSHAP | 0.04 | 0.19 | 0.17 | -0.06 | -0.10 | 0.21 | 0.14 | -0.34 | -0.19 | 0.09 | 0.08 |
| | | | | | | **RBO** | | | | | |
| LIME | 0.54 | 0.66 | 0.49 | 0.63 | 0.55 | 0.56 | 0.41 | 0.59 | 0.66 | 0.68 | 0.39 |
| TreeSHAP | 0.49 | 0.67 | 0.55 | 0.66 | 0.59 | 0.52 | 0.49 | 0.61 | 0.67 | 0.63 | 0.44 |
| KernelSHAP | 0.57 | 0.69 | 0.56 | 0.63 | 0.57 | 0.55 | 0.56 | 0.61 | 0.68 | 0.64 | 0.45 |

considered datasets are randomly split into 80% training and and 20% test data. For images, 15 test instances are randomly selected in each test set for explanation while all tabular test instances are explained. For all datasets, gradient boosted trees (BTs) are trained by XGBoost (Chen and Guestrin, 2016), where each BT consists of 25 trees of depth 3 per class.[5]

## 5.1 EXACT FORMAL FEATURE ATTRIBUTION

Here we consider examples where we can compute the *exact* FFA values by computing all AXp's. To compare FFA with feature attribution produced by LIME, TreeSHAP and KernelSHAP, we take the absolute values of their feature attribution and normalize the values into $[0, 1]$. The *error* is measured as Manhattan distance, i.e. the sum of absolute differences across all features. We also compare feature rankings according to the competitors (again using absolute values for LIME, TreeSHAP and KernelSHAP) using Kendall's Tau (Kendall, 1938) and rank-biased overlap (RBO) (Webber et al., 2010) metrics.[6] Kendall's Tau and RBO are measured on a scale $[-1, 1]$ and $[0, 1]$, respectively. A higher value in both metrics indicates better agreement or closeness between a ranking and FFA.

**Tabular Data.** Figure 3 exemplifies a comparison of FFA, LIME, TreeSHAP and KernelSHAP on a randomly selected instance of the Compas dataset (Angwin et al., 2016). While FFA and LIME agree on the most important feature, "Asian", TreeSHAP gives it very little weight. None of LIME, TreeSHAP and KernelSHAP agree with FFA, though there is clearly some similarity.

Table 1 details the comparison conducted on 11 tabular datasets, including *adult*, *compas*, and *recidivism* datasets commonly used in XAI. For each dataset, we calculate the metric for each individual instance and then average the outcomes to obtain the final result for that dataset. As can be observed, the errors of LIME's feature attribution across these datasets span from 1.39 to 5.13. TreeSHAP and KernelSHAP demonstrate errors within a range $[1.40, 4.76]$ and $[0.83, 4.81]$, respectively. These three approaches also exhibit comparable performance in relation to the two ranking comparison metrics. The values of Kendall's Tau for LIME (resp. TreeSHAP and KernelSHAP) range from $-0.36$ to $0.22$ (resp. $-0.39$ to $0.27$ and $-0.34$ to $0.21$). Regarding the RBO values, LIME exhibits values between

---

[5]Test accuracy for MNIST digits is 0.99, while it is 0.83 for PneumoniaMNIST. This holds both for the 28 × 28 and 10 × 10 versions of the datasets. The average accuracy across the 11 selected tabular datasets is 0.80.

[6]Kendall's Tau and RBO measure the similarity between two ranked lists. The former is a correlation coefficient assessing the ordinal association while the latter accounts for the order and the depth of the overlap.

Table 2: Comparison on $10 \times 10$ Images of FFA versus competitors and FFA approximations.

| Dataset
($|\mathcal{F}| = 100$) | LIME | TreeSHAP | KernelSHAP | FFA$_{10}$ | FFA$_{30}$ | FFA$_{60}$ | FFA$_{120}$ | FFA$_{600}$ | FFA$_{1200}$ |
|---|---|---|---|---|---|---|---|---|---|
| | | | | Error | | | | | |
| 10×10-mnist-1vs3 | 11.50 | 10.07 | 10.43 | 5.74 | 5.33 | 4.97 | 4.62 | 3.37 | 2.67 |
| 10×10-mnist-1vs7 | 12.64 | 8.28 | 8.27 | 4.16 | 3.58 | 2.94 | 2.50 | 1.42 | 1.01 |
| 10×10-pneumoniamnist | 17.32 | 17.90 | 18.15 | 5.37 | 4.32 | 3.78 | 3.39 | 2.22 | 1.64 |
| | | | | Kendall's Tau | | | | | |
| 10×10-mnist-1vs3 | -0.15 | 0.48 | 0.14 | 0.49 | 0.57 | 0.62 | 0.65 | 0.74 | 0.80 |
| 10×10-mnist-1vs7 | -0.33 | 0.47 | 0.17 | 0.52 | 0.63 | 0.70 | 0.77 | 0.85 | 0.89 |
| 10×10-pneumoniamnist | -0.02 | 0.24 | 0.01 | 0.58 | 0.71 | 0.79 | 0.80 | 0.89 | 0.92 |
| | | | | RBO | | | | | |
| 10×10-mnist-1vs3 | 0.20 | 0.50 | 0.53 | 0.61 | 0.65 | 0.69 | 0.74 | 0.81 | 0.84 |
| 10×10-mnist-1vs7 | 0.19 | 0.58 | 0.53 | 0.73 | 0.77 | 0.81 | 0.86 | 0.90 | 0.90 |
| 10×10-pneumoniamnist | 0.21 | 0.37 | 0.47 | 0.61 | 0.70 | 0.73 | 0.77 | 0.83 | 0.87 |

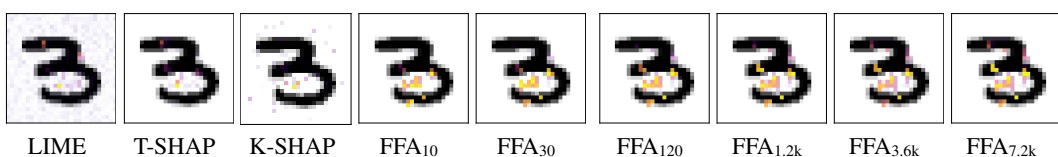

| LIME | T-SHAP | K-SHAP | FFA$_{10}$ | FFA$_{30}$ | FFA$_{120}$ | FFA$_{1.2k}$ | FFA$_{3.6k}$ | FFA$_{7.2k}$ |

Figure 4: $28 \times 28$ MNIST 1 vs. 3. The prediction is digit 3. The *plasma* gradient is used ranging from deep purple for the least important features to vibrant yellow for the most important features.

Table 3: Comparison on $28 \times 28$ Images of FFA$_{7200}$ versus competitors and FFA approximations.

| Dataset
($|\mathcal{F}| = 784$) | LIME | TreeSHAP | KernelSHAP | FFA$_{10}$ | FFA$_{30}$ | FFA$_{120}$ | FFA$_{1200}$ | FFA$_{3600}$ |
|---|---|---|---|---|---|---|---|---|
| | | | | Error | | | | |
| 28×28-mnist-1vs3 | 49.66 | 22.77 | 26.42 | 9.44 | 7.61 | 6.81 | 3.13 | 2.69 |
| 28×28-mnist-1vs7 | 55.10 | 24.92 | 28.93 | 11.78 | 9.58 | 6.94 | 3.30 | 2.18 |
| 28×28-pneumoniamnist | 62.94 | 31.55 | 41.92 | 8.17 | 7.81 | 5.69 | 3.77 | 3.10 |
| | | | | Kendall's Tau | | | | |
| 28×28-mnist-1vs3 | -0.80 | 0.42 | -0.40 | 0.44 | 0.62 | 0.69 | 0.86 | 0.87 |
| 28×28-mnist-1vs7 | -0.79 | 0.34 | -0.53 | 0.40 | 0.56 | 0.72 | 0.87 | 0.92 |
| 28×28-pneumoniamnist | -0.66 | 0.24 | -0.65 | 0.34 | 0.50 | 0.67 | 0.80 | 0.87 |
| | | | | RBO | | | | |
| 28×28-mnist-1vs3 | 0.03 | 0.40 | 0.32 | 0.43 | 0.50 | 0.61 | 0.83 | 0.88 |
| 28×28-mnist-1vs7 | 0.03 | 0.34 | 0.30 | 0.40 | 0.45 | 0.58 | 0.83 | 0.93 |
| 28×28-pneumoniamnist | 0.03 | 0.23 | 0.25 | 0.31 | 0.35 | 0.42 | 0.66 | 0.83 |

0.39 and 0.68, whereas TreeSHAP demonstrates values ranging from 0.44 to 0.67 and KernelSHAP yields results between 0.45 and 0.69. Overall, as Table 1 indicates, neither LIME, nor TreeSHAP, nor KernelSHAP agree with FFA and so neither of them capture feature relevancy well enough.

**$10 \times 10$ Digits.** We now compare the results on $10 \times 10$ downscaled MNIST digits and Pneumoni-aMNIST images, where it is feasible to compute all AXp's. Table 2 compares LIME's, TreeSHAP's, KernelSHAP's feature attribution and approximate FFA. Here, we run AXp enumeration for a number of seconds, which is denoted as FFA$_*$, $* \in \mathbb{R}^+$. The runtime required for each image by LIME and TreeSHAP is less than one second, whereas KernelSHAP takes 33.26s per image on average. The results show that the errors of our approximation are small, even after 10 seconds it beats LIME, TreeSHAP and KernelSHAP, and decreases as we generate more AXp's. The results for the orderings show again that after 10 seconds, FFA$_*$ ordering gets closer to the exact FFA than LIME, TreeSHAP and KernelSHAP. Observe how LIME is particularly far away from the *exact* FFA ordering.

**Summary.** *Exact FFA may be efficiently approximated without exhaustively computing all AXp's. While feature attribution determined by LIME, TreeSHAP and KernelSHAP is not meant to approximate FFA, the observed disagreement demonstrates that they fail to capture the true feature relevancy and so may be unable to provide a human-decision maker with useful insights, despite being fast.*

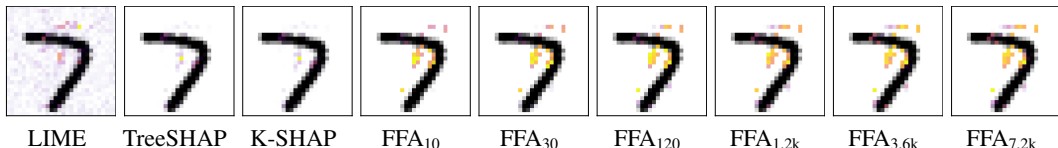

Figure 5: $28 \times 28$ MNIST 1 vs. 7. The prediction is digit 7.

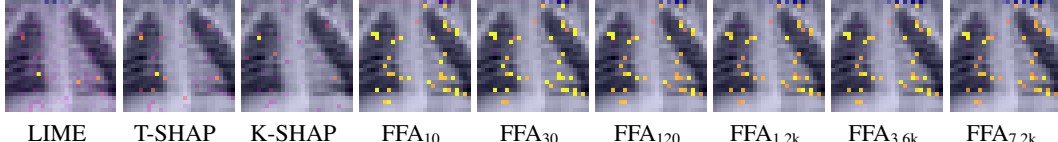

Figure 6: $28 \times 28$ PneumoniaMNIST. The prediction is normal.

## 5.2 APPROXIMATING FORMAL FEATURE ATTRIBUTION

Since the problem of formal feature attribution is computationally expensive to solve, it is not surprising that computing FFA may be challenging in practice. Table 2 suggests that our approach gets good FFA approximations even if we only collect AXp's for a short time. Here we compare the fidelity of our approach versus the approximate FFA computed after 2 hours (7200s). Figures 4, 5, and 6 depict feature attribution generated by LIME, TreeSHAP, KernelSHAP and $FFA_*$ for the three selected $28 \times 28$ images. The comparison between LIME, TreeSHAP, KernelSHAP and the approximate FFA computation is detailed in Table 3. The LIME and TreeSHAP processing time for each image is less than one second, where KernelSHAP requires 98.56s to process on average. The average findings detailed in Table 3 are consistent with those shown in Table 2. Namely, FFA approximation yields better errors, Kendall's Tau and RBO values, outperforming LIME, TreeSHAP and KernelSHAP after 10 seconds. Furthermore, the results demonstrate that after 10 seconds our approach places feature attribution closer to $FFA_{7200}$ compared to LIME, TreeSHAP and KernelSHAP hinting on the features that are truly relevant for the prediction.

## 6 LIMITATIONS

Despite the rigorous guarantees provided by formal feature attribution and high-quality of the result explanations, the following limitations can be identified. First, our approach relies on formal reasoning and thus requires an ML model of interest to admit a representation in some fragments of first-order logic, and the corresponding reasoner to deal with it (Marques-Silva and Ignatiev, 2022). Second, the complexity of computing exact FFA demands the development of effective methods of FFA approximation. Finally, though our experimental evidence suggests that FFA approximations quickly converge to the exact values of FFA, whether or not this holds in general remains an open question.

## 7 CONCLUSIONS

Most approaches to XAI are heuristic methods that are susceptible to unsoundness and out-of-distribution sampling. Formal approaches to XAI have so far concentrated on the problem of feature selection, detecting which features are important for justifying a classification decision, and not on feature attribution, where we can understand the weight of a feature in making such a decision. In this paper we define the first formal approach to feature attribution (FFA) we are aware of, using the proportion of abductive explanations in which a feature occurs to weight its importance. We show that we can compute FFA exactly for many classification problems, and when we cannot we can compute effective approximations. Existing heuristic approaches to feature attribution do not agree with FFA. Sometimes they markedly differ, for example, assigning no weight to a feature that appears in (a large number of) explanations, or assigning (large) non-zero weight to a feature that is irrelevant for the prediction. Overall, the paper argues that if we agree that FFA is a correct measure of feature attribution then we need to investigate methods that compute good FFA approximations quickly.

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

# Appendices

These supplementary materials repeat the above experiments for weighted formal feature attribution and detail the corresponding experimental results. They also demonstrate the use of FFA and WFFA in the case of CIFAR-10 images as well as showcase the use of both unweighted and weighted formal feature attribution in a real-world scenario of Just-in-Time (JIT) defect prediction (Pornprasit et al., 2021) where post-hoc explanations are saught.

## A  EXACT WEIGHTED FORMAL FEATURE ATTRIBUTION

In this appendix, we once again limit our analysis to instances where we can calculate the *exact* WFFA values for the instance of interest by enumerating all AXp's. Also, the settings used in Section 5 are applied here, i.e. we take the absolute values of feature attribution assigned by LIME (Ribeiro et al., 2016), TreeSHAP (Lundberg et al., 2020) and KernelSHAP (Lundberg and Lee, 2017), and normalize them within the range of $[0, 1]$. Just like in the main text of the paper, we then compare these approaches with normalized WFFA values in terms of errors, Kendall's Tau (Kendall, 1938) and rank-biased overlap (RBO) (Webber et al., 2010).

### A.1  TABULAR DATA

A comparison of WFFA, LIME, TreeSHAP and KernelSHAP on a randomly selected instance (for illustrativity, the instance is the same as the one shown earlier in Figure 3) of the Compas dataset (Angwin et al., 2016) is exemplified in Figure 7. We can observe the patterns similar to those depicted in Figure 3. The feature that WFFA considers most important is "Asian" while this viewpoint is shared by LIME but disputed by TreeSHAP and KernelSHAP. However, none of LIME, TreeSHAP and KernelSHAP fully align with WFFA, although there is evident similarity between them. As with FFA, these observations can be generalized to the other instances of Compas, as discussed below.

Table 4 presents a comparison of WFFA against LIME, SHAP and KernelSHAP on the 11 selected tabular datasets as in Table 1, demonstrating similarities in the findings observed for WFFA and FFA for these datasets. The average runtime for generating the exact WFFA in a dataset varies between 0.18 and 1.89 seconds while the average number of AXp's per instance to explain and so to compute exact WFFA in a dataset ranges from 1.40 to 33.33. LIME exhibits errors ranging from 1.37 to 4.96 across these datasets while TreeSHAP shows similar errors spanning from 1.36 to 4.67 and KernelSHAP displays errors between 0.79 and 4.42. Besides errors, LIME, TreeSHAP and KernelSHAP yield comparable outcomes in terms of the two ranking comparison metrics. The values of Kendall's Tau for LIME are between $-0.35$ and $0.25$, whereas the values for TreeSHAP and KernelSHAP range from $-0.38$ to $0.31$ and $-0.33$ to $0.31$, respectively. Regarding RBO values, LIME (resp. TreeSHAP and KernelSHAP) demonstrates values ranging from 0.38 to 0.69 (resp. 0.43 to 0.67 and 0.43 to 0.71). Overall and consistent with the FFA findings shown earlier in Table 1, Table 4 indicates that LIME, TreeSHAP and KernelSHAP fail to achieve close enough agreement with WFFA.

### A.2  $10 \times 10$ DIGITS

Table 5 provides a comprehensive comparison of approximate WFFA against feature attribution reported by LIME, TreeSHAP and KernelSHAP with respect to the exact WFFA values, conducted on the downscaled MNIST digists and PneumoniaMNIST images, where exhaustive AXp enumeration is feasible. The values of feature attribution generated by LIME, TreeSHAP, KernelSHAP and approximate WFFA$_*$ for the three selected $10 \times 10$ images are shown in Figure 11, Figure 12, and Figure 13. Over time, the number of features included in the AXp's increases, and the weighted attribution of each feature changes converging to the exact WFFA. The results shown in Figure 8, Figure 9, and Figure 10 align with the main finding for FFA approximation shown earlier. Furthermore, the results shown in Table 5 are also consistent with FFA observations in Table 2. Both LIME and TreeSHAP can process each image within a runtime of less than one second, while KernelSHAP takes 33.26s per image on average. The average runtime and average number of AXp's generated for $10 \times 10$ MNIST 1 vs 3 (resp. 1 vs 7) are 14264.78s and 15781.87 (resp. 6834.61s and 4028.27), while the

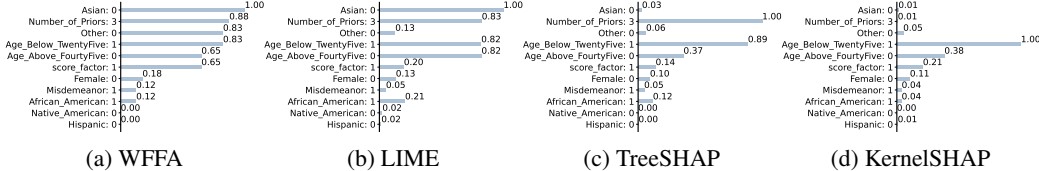

(a) WFFA      (b) LIME      (c) TreeSHAP      (d) KernelSHAP

Figure 7: Explanations for an instance of Compas $\mathbf{v} = \{\#\text{Priors} = 3, \text{Score\_factor} = 1, \text{Age\_Above\_FourtyFive} = 0, \text{Age\_Below\_TwentyFive} = 1, \text{African\_American} = 1, \text{Asian} = 0, \text{Hispanic} = 0, \text{Native\_American} = 0, \text{Other} = 0, \text{Female} = 0, \text{Misdemeanor} = 1\}$ predicted as Two\_yr\_Recidivism = true.

Table 4: LIME, TreeSHAP and KernelSHAP versus WFFA on tabular data.

| Dataset | adult | appendicitis | australian | cars | compas | heart-statlog | hungarian | lending | liver-disorder | pima | recidivism |
|---|---|---|---|---|---|---|---|---|---|---|---|
| $|\mathcal{F}|$ | (12) | (7) | (14) | (8) | (11) | (13) | (13) | (9) | (6) | (8) | (15) |
| **Approach** | | | | | | **Error** | | | | | |
| LIME | 4.32 | 2.06 | 4.96 | 1.48 | 3.26 | 4.40 | 4.43 | 1.37 | 2.37 | 2.63 | 4.66 |
| TreeSHAP | 4.29 | 1.87 | 4.31 | 1.36 | 2.63 | 3.61 | 4.00 | 1.43 | 2.25 | 2.91 | 4.67 |
| KernelSHAP | 4.14 | 1.90 | 4.42 | 0.79 | 2.54 | 3.43 | 3.87 | 1.33 | 2.20 | 2.86 | 4.72 |
| | | | | | | **Kendall's Tau** | | | | | |
| LIME | 0.11 | 0.17 | 0.25 | -0.08 | -0.08 | 0.22 | 0.08 | -0.35 | -0.17 | 0.25 | 0.08 |
| TreeSHAP | 0.07 | 0.23 | 0.31 | -0.07 | -0.07 | 0.22 | 0.26 | -0.38 | -0.16 | 0.15 | 0.16 |
| KernelSHAP | 0.08 | 0.31 | 0.20 | -0.03 | -0.07 | 0.25 | 0.21 | -0.33 | -0.14 | 0.16 | 0.11 |
| | | | | | | **RBO** | | | | | |
| LIME | 0.53 | 0.65 | 0.48 | 0.64 | 0.56 | 0.56 | 0.40 | 0.59 | 0.65 | 0.69 | 0.38 |
| TreeSHAP | 0.48 | 0.67 | 0.55 | 0.66 | 0.59 | 0.52 | 0.49 | 0.61 | 0.67 | 0.64 | 0.43 |
| KernelSHAP | 0.55 | 0.71 | 0.54 | 0.63 | 0.57 | 0.53 | 0.54 | 0.61 | 0.68 | 0.64 | 0.43 |

values in $10 \times 10$ PneumoniaMNIST are 8656.18s and 8802.87, respectively. Similarly to the results in Table 2, Table 5 indicates that our approximation yields small errors. Even after 10 seconds, it outperforms LIME, TreeSHAP and KernelSHAP, and the errors continue to decrease as we compute more AXp's. Once again, the results of the orderings demonstrate that after 10 seconds, the ordering of WFFA$_*$ approaches closer to the exact WFFA compared to LIME, TreeSHAP and KernelSHAP, and converges to the exact WFFA ordering with the growth of the number AXp's enumerated. As can also be seen, LIME exhibits a substantial distance from the *exact* WFFA ordering.

### A.3 SUMMARY

The findings of this section again indicate that we can confidently obtain valuable approximations of the exact WFFA values without the need to exhaustively enumerate all AXp's for a given data instance. Similarly to the main results shown in Section 5, it is once again worth noting that feature attribution determined by LIME, TreeSHAP and KernelSHAP is inconsistent with WFFA and its approximations, and hence does not capture formal feature relevancy, despite being computationally fast.

## B APPROXIMATE WEIGHTED FORMAL FEATURE ATTRIBUTION

As argued in Section 3, the exact WFFA computation can be difficult in practice, due to the complexity of the problem. But as Table 5 indicates, our approach can yield decent WFFA approximations even with a short duration of collecting AXp's. Here we assess the fidelity of our approach in contrast to the approximate WFFA computed after a duration of 2 hours (7200s). WFFA$_*$ and the values of feature attribution generated by LIME, TreeSHAP and KernelSHAP for the three considered 28 $\times$ 28 images are depicted in Figure 14, 15, and 16. As time progresses, the accumulated AXp's incorporate an increasing number of features, and as a result the value of weighted attribution for each feature can change. Table 6 details the comparison between LIME, TreeSHAP, KernelSHAP and the approximate WFFA. Both LIME and TreeSHAP process each image in under one second, whereas KernelSHAP, on average, requires 98.56s to process. The average results presented in Table 6 are

Table 5: Comparison on $10 \times 10$ Images of WFFA versus LIME, TreeSHAP, KernelSHAP and WFFA approximations.

| Dataset | LIME | TreeSHAP | KernelSHAP | WFFA$_{10}$ | WFFA$_{30}$ | WFFA$_{60}$ | WFFA$_{120}$ | WFFA$_{1200}$ |
|---|---|---|---|---|---|---|---|---|
| $\lvert\mathcal{F}\rvert = 100$ | | | | Error | | | | |
| 10×10-mnist-1vs3 | 11.28 | 9.81 | 10.15 | 5.52 | 5.12 | 4.83 | 3.32 | 2.61 |
| 10×10-mnist-1vs7 | 12.46 | 8.11 | 8.09 | 4.07 | 3.47 | 2.83 | 1.34 | 0.97 |
| 10×10-pneumoniamnist | 17.25 | 17.84 | 18.08 | 5.33 | 4.29 | 3.76 | 2.20 | 1.63 |
| | | | | Kendall's Tau | | | | |
| 10×10-mnist-1vs3 | -0.14 | 0.48 | 0.15 | 0.53 | 0.60 | 0.64 | 0.75 | 0.81 |
| 10×10-mnist-1vs7 | -0.33 | 0.47 | 0.17 | 0.58 | 0.65 | 0.73 | 0.86 | 0.90 |
| 10×10-pneumoniamnist | -0.02 | 0.24 | 0.01 | 0.67 | 0.74 | 0.80 | 0.90 | 0.92 |
| | | | | RBO | | | | |
| 10×10-mnist-1vs3 | 0.20 | 0.50 | 0.53 | 0.63 | 0.67 | 0.70 | 0.81 | 0.84 |
| 10×10-mnist-1vs7 | 0.19 | 0.58 | 0.53 | 0.73 | 0.77 | 0.81 | 0.90 | 0.91 |
| 10×10-pneumoniamnist | 0.21 | 0.37 | 0.46 | 0.63 | 0.70 | 0.74 | 0.82 | 0.87 |

| LIME | T-SHAP | K-SHAP | FFA$_{10}$ | FFA$_{30}$ | FFA$_{60}$ | FFA$_{120}$ | FFA$_{1.2k}$ | FFA |

Figure 8: $10 \times 10$ MNIST 1 vs. 3. Competitors and FFA$_*$. The prediction is 3.

Table 6: Comparison on $28 \times 28$ Images of WFFA$_{7.2k}$ versus LIME, TreeSHAP , KernelSHAP and WFFA approximations.

| Dataset | LIME | TreeSHAP | KernelSHAP | WFFA$_{10}$ | WFFA$_{30}$ | WFFA$_{120}$ | WFFA$_{1200}$ | WFFA$_{3600}$ |
|---|---|---|---|---|---|---|---|---|
| $\lvert\mathcal{F}\rvert = 784$ | | | | Error | | | | |
| 28,28-mnist-1,3 | 49.28 | 22.33 | 25.97 | 9.22 | 7.50 | 6.69 | 3.08 | 2.75 |
| 28,28-mnist-1,7 | 54.78 | 24.39 | 28.39 | 11.53 | 9.40 | 7.00 | 3.33 | 2.29 |
| 28,28-pneumoniamnist | 62.88 | 31.46 | 41.80 | 8.17 | 7.74 | 5.67 | 3.75 | 3.08 |
| | | | | Kendall's Tau | | | | |
| 28,28-mnist-1,3 | -0.80 | 0.42 | -0.40 | 0.49 | 0.64 | 0.70 | 0.86 | 0.88 |
| 28,28-mnist-1,7 | -0.79 | 0.34 | -0.53 | 0.43 | 0.57 | 0.72 | 0.87 | 0.92 |
| 28,28-pneumoniamnist | -0.66 | 0.24 | -0.65 | 0.37 | 0.57 | 0.69 | 0.81 | 0.88 |
| | | | | RBO | | | | |
| 28,28-mnist-1,3 | 0.03 | 0.40 | 0.32 | 0.45 | 0.54 | 0.63 | 0.84 | 0.89 |
| 28,28-mnist-1,7 | 0.03 | 0.34 | 0.29 | 0.41 | 0.47 | 0.60 | 0.81 | 0.91 |
| 28,28-pneumoniamnist | 0.03 | 0.23 | 0.24 | 0.30 | 0.35 | 0.43 | 0.65 | 0.81 |

Table 7: Just-in-time Defect Prediction comparison of WFFA versus LIME, TreeSHAP and KernelSHAP.

| Approach | openstack ($\lvert\mathcal{F}\rvert = 13$) | | | qt ($\lvert\mathcal{F}\rvert = 16$) | | |
|---|---|---|---|---|---|---|
| | Error | Kendall's Tau | RBO | Error | Kendall's Tau | RBO |
| LIME | 4.79 | 0.08 | 0.56 | 5.60 | -0.07 | 0.45 |
| TreeSHAP | 5.01 | 0.02 | 0.54 | 5.17 | -0.11 | 0.44 |
| KernelSHAP | 5.06 | -0.33 | 0.64 | 5.19 | -0.44 | 0.69 |

consistent with those illustrated in Table 5 and the FFA results depicted in Table 2 and Table 3. Table 6 demonstrates that after only 10 seconds, our WFFA approximation outperforms both LIME, TreeSHAP and KernelSHAP in terms of errors, Kendall's Tau, and RBO values. Additionally, after 10 seconds our approach produces weighted feature attributions, which is closer to WFFA$_{7200}$ compared to LIME, TreeSHAP and KernelSHAP. This suggests that our approach effectively identifies the

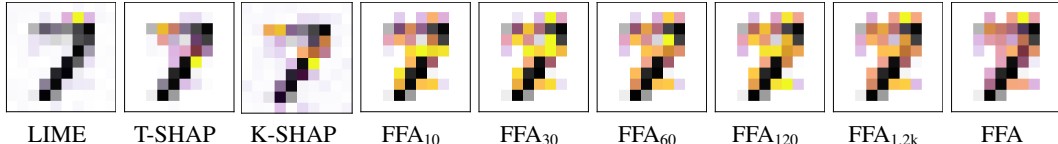

| LIME | T-SHAP | K-SHAP | FFA$_{10}$ | FFA$_{30}$ | FFA$_{60}$ | FFA$_{120}$ | FFA$_{1.2k}$ | FFA |

Figure 9: $10 \times 10$ MNIST 1 vs. 7. Competitors and FFA$_*$. The prediction is 7.

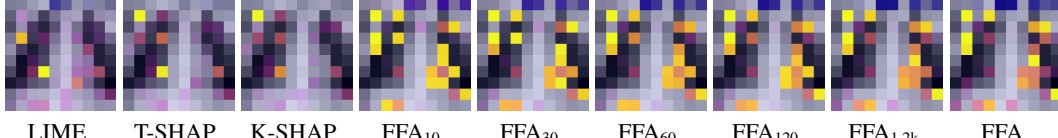

| LIME | T-SHAP | K-SHAP | FFA$_{10}$ | FFA$_{30}$ | FFA$_{60}$ | FFA$_{120}$ | FFA$_{1.2k}$ | FFA |

Figure 10: $10 \times 10$ PneumoniaMNIST. Competitors and FFA$_*$. The prediction is pneumonia.

Table 8: JIT Defect Prediction comparison of FFA versus LIME, TreeSHAP and KernelSHAP.

| Approach | openstack ($|\mathcal{F}| = 13$) | | | qt ($|\mathcal{F}| = 16$) | | |
|---|---|---|---|---|---|---|
| | Error | Kendall's Tau | RBO | Error | Kendall's Tau | RBO |
| LIME | 4.84 | 0.05 | 0.55 | 5.63 | -0.08 | 0.45 |
| TreeSHAP | 5.08 | 0.00 | 0.53 | 5.22 | -0.13 | 0.44 |
| KernelSHAP | 5.06 | -0.33 | 0.64 | 5.19 | -0.44 | 0.69 |

features that are genuinely relevant for the prediction, which is in stark contrast to LIME, TreeSHAP and KernelSHAP.

## C   APPLICATION IN JUST-IN-TIME DEFECT PREDICTION

Modern software companies often engage in the rapid and frequent release of software products in short cycles. Because of the exponential growth of highly complex source code, such rapid-release software development presents significant challenges for under-resourced Software Quality Assurance (SQA) teams. Developers are unable to thoroughly ensure the highest quality of all newly developed code commits or pull requests within the limited time and resources available, due to the time-consuming and costly nature of various SQA activities, e.g. code review. To address this issue, a recent approach called Just-in-Time (JIT) defect prediction (Kim et al., 2007; Kamei et al., 2013; Pornprasit and Tantithamthavorn, 2021; Lin et al., 2021) has been proposed. This approach aims to predict whether a commit will introduce software defects in the future such that development teams can prioritize their limited SQA resources on the riskiest commits or pull requests.

However, the JIT defect prediction approach has frequently been criticized for being opaque and lacking explainability for practitioners. Model-agnostic explainability methods, e.g. LIME, Tree-SHAP and KernelSHAP, fail to respect the actual feature relevancy and so can hardly guarantee accurate feature attribution, as discussed earlier in this appendix and Section 5). Here, we apply the computation of both FFA and WFFA in the setting of JIT defection prediction and demonstrate that they can represent a viable approach to addressing practical explainability challenges.

In particular, where we use logistic regression models built on two widely-used large-scale open-source datasets, namely Openstack and Qt, which are commonly used in JIT defect prediction studies (Pornprasit et al., 2021). The property of monotonicity in logistic regression allows us to enumerate explanations efficiently, following the approach of (Marques-Silva et al., 2021). By leveraging this method, we can enumerate all abductive explanation for each instance *within one second* and hence compute both *exact FFA* and *exact WFFA*. Table 8 details the comparison between exact FFA values and feature attribution by the other competitors in terms of the three selected metrics while the comparison of WFFA, LIME, TreeSHAP and KernelSHAP is provided in Table 7. Similar to all the other findings above (see Table 1, Table 2, Table 3, Table 4, Table 5, and Table 6), LIME, TreeSHAP and KernelSHAP misalign with unweighted and weighted formal feature attribution, although there are some similarities between them.

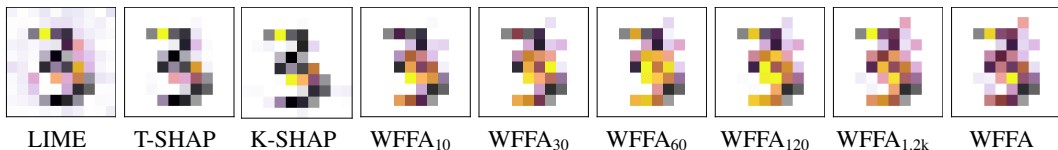

| LIME | T-SHAP | K-SHAP | WFFA$_{10}$ | WFFA$_{30}$ | WFFA$_{60}$ | WFFA$_{120}$ | WFFA$_{1.2k}$ | WFFA |

Figure 11: $10 \times 10$ MNIST 1 vs. 3. Competitors and WFFA$_*$. The prediction is 3.

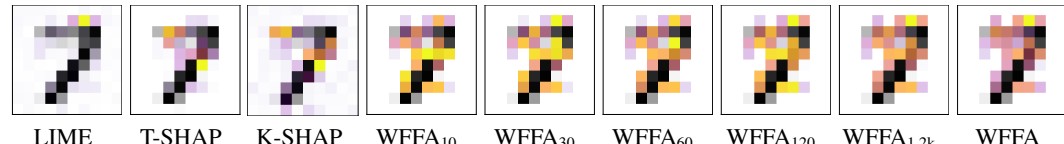

| LIME | T-SHAP | K-SHAP | WFFA$_{10}$ | WFFA$_{30}$ | WFFA$_{60}$ | WFFA$_{120}$ | WFFA$_{1.2k}$ | WFFA |

Figure 12: $10 \times 10$ MNIST 1 vs. 7. Competitors and WFFA$_*$. The prediction is 7.

Table 9: Comparison on CIFAR Images of FFA$_{7200}$ versus LIME, TreeSHAP and KernelSHAP and FFA approximations.

| Dataset | LIME | TreeSHAP | KernelSHAP | FFA$_{30}$ | FFA$_{60}$ | FFA$_{120}$ | FFA$_{600}$ | FFA$_{1200}$ | FFA$_{3600}$ |
|---|---|---|---|---|---|---|---|---|---|
| $(\lvert\mathcal{F}\rvert = 1024)$ | | | | Error | | | | | |
| | 222.15 | 104.22 | 89.33 | 40.14 | 7.06 | 5.22 | 3.48 | 3.21 | 1.76 |
| 32,32-cifar-10-ship,truck | | | | Kendall's Tau | | | | | |
| | -0.75 | -0.75 | -0.81 | -0.28 | 0.25 | 0.37 | 0.65 | 0.71 | 0.85 |
| | | | | RBO | | | | | |
| | 0.02 | 0.16 | 0.39 | 0.43 | 0.51 | 0.55 | 0.69 | 0.74 | 0.84 |

Table 10: Comparison on CIFAR Images of WFFA$_{7200}$ versus LIME, TreeSHAP and KernelSHAP and WFFA approximations.

| Dataset | LIME | TreeSHAP | KernelSHAP | FFA$_{30}$ | FFA$_{60}$ | FFA$_{120}$ | FFA$_{600}$ | FFA$_{1200}$ | FFA$_{3600}$ |
|---|---|---|---|---|---|---|---|---|---|
| $(\lvert\mathcal{F}\rvert = 1024)$ | | | | Error | | | | | |
| | 222.14 | 104.19 | 89.29 | 40.12 | 7.05 | 5.21 | 3.45 | 3.19 | 1.76 |
| 32,32-cifar-10-ship,truck | | | | Kendall's Tau | | | | | |
| | -0.75 | -0.75 | -0.91 | -0.39 | 0.17 | 0.35 | 0.75 | 0.79 | 0.88 |
| | | | | RBO | | | | | |
| | 0.02 | 0.11 | 0.27 | 0.24 | 0.35 | 0.41 | 0.60 | 0.65 | 0.76 |

# D  CIFAR-10 IMAGES

The appendix presents the results of the well-known CIFAR-10 image dataset, which consists of $32 \times 32$ color images in 10 classes. We create a concrete binary classification task within this dataset: *ship* vs. *truck*, where 5000 and 1000 images for each class are included in the training and test data, respectively. For this dataset, 15 test instances are randomly selected in the test set for evaluation. We applied XGBoost (Chen and Guestrin, 2016) to train gradient boosted trees (BTs) on this dataset, where each BT consists of 50 trees of maximum depth of 3 per class and the test accuracy is 0.87. As discussed in Section 5.2, it is not surprising that computation of exact FFA/WFFA may be challenging in practice, as the problem of formal (weighted) feature attribution "lives" in $\Sigma_2^P$. Table 2 and Table 5 demonstrate that our approach yields good FFA/WFFA approximations even when we only collect AXp's for a short time. In this appendix, we compare the fidelity of our approach with the approximate FFA and WFFA computed after 2 hours (7200s).

The comparisons between LIME, TreeSHAP, KernelSHAP and the approximate FFA/WFFA computation are detailed in Table 9 and Table 10. Figures 17 and 18 present feature attributions generated by LIME, TreeSHAP, KernelSHAP, FFA$_*$ and WFFA$_*$ for the dataset. The TreeSHAP processing time for each image is less than one second, where LIME and KernelSHAP 5.50s and 188.98s to process on average. The average findings detailed in Figures 17 and 18 align with those presented in

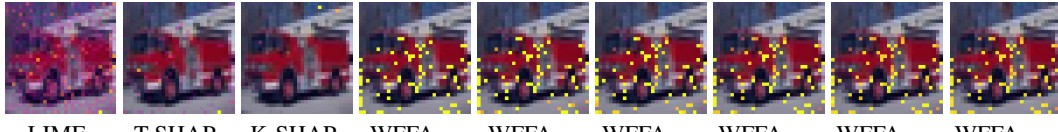

LIME    T-SHAP    K-SHAP    WFFA$_{10}$    WFFA$_{30}$    WFFA$_{60}$    WFFA$_{120}$    WFFA$_{1.2k}$    WFFA

Figure 13: $10 \times 10$ PneumoniaMNIST. Competitors and WFFA$_*$. The prediction is pneumonia.

LIME    T-SHAP    K-SHAP    WFFA$_{10}$    WFFA$_{30}$    WFFA$_{120}$    WFFA$_{1.2k}$    WFFA$_{3.6k}$    WFFA$_{7.2k}$

Figure 14: $28 \times 28$ MNIST 1 vs. 3. Competitors and WFFA$_*$. The prediction is digit 3.

LIME    T-SHAP    K-SHAP    WFFA$_{10}$    WFFA$_{30}$    WFFA$_{120}$    WFFA$_{1.2k}$    WFFA$_{3.6k}$    WFFA$_{7.2k}$

Figure 15: $28 \times 28$ MNIST 1 vs. 7. Competitors and WFFA$_*$. The prediction is digit 7.

LIME    T-SHAP    K-SHAP    WFFA$_{10}$    WFFA$_{30}$    WFFA$_{120}$    WFFA$_{1.2k}$    WFFA$_{3.6k}$    WFFA$_{7.2k}$

Figure 16: $28 \times 28$ PneumoniaMNIST. Competitors and WFFA$_*$. The prediction is normal.

LIME    T-SHAP    K-SHAP    FFA$_{30}$    FFA$_{60}$    FFA$_{120}$    FFA$_{1.2k}$    FFA$_{3.6k}$    FFA$_{7.2k}$

Figure 17: $32 \times 32$ CIFAR Ship vs. Truck. Competitors and FFA$_*$. The prediction is truck.

LIME    T-SHAP    K-SHAP    WFFA$_{30}$    WFFA$_{60}$    WFFA$_{120}$    WFFA$_{1.2k}$    WFFA$_{3.6k}$    WFFA$_{7.2k}$

Figure 18: $32 \times 32$ CIFAR Ship vs. Truck. Competitors and WFFA$_*$. The prediction is truck.

Figures 2, 3, 5 and 6. Namely, in terms of (unweighted) FFA, its approximation yields superior errors, Kendall's Tau and RBO values, surpassing LIME, TreeSHAP and KernelSHAP after 30 seconds. This observation also holds for WFFA approximation. Furthermore, the results demonstrate that after 30 seconds our approach provides feature attributions closer to FFA$_{7200}$ and WFFA$_{7200}$ compared with LIME, TreeSHAP and KernelSHAP, indicating the features that are truly relevant for the prediction.

