# OpenReview forum: "On Formal Feature Attribution and Its Approximation"
_ICLR.cc/2024/Conference — ICLR 2024 Conference Withdrawn Submission_

### Official Review · Reviewer_2iyx · 2023-10-29

**Soundness:** 1 poor
**Presentation:** 2 fair
**Contribution:** 2 fair
**Rating:** 5
**Confidence:** 3

**Summary:**

The authors of this study have outlined the shortcomings of current XAI algorithms and have put forward a new concept called Formal Feature Attribution (FFA) that offers more reliable results and can be approximated more efficiently. They have compared the results obtained from FFA with other state-of-the-art algorithms and have shown that their FFA approximation algorithm has unique properties that set it apart from the rest.

**Strengths:**

The paper did a good job of introducing the two important properties in XAI, abductive explanations (AXp's) and contrastive explanations (CXp's), and the attractive properties of having these two properties satisfied. The authors then proposed to use the scheme of formal feature attribution (FFA) to address these two properties.

To address the computational challenge of FFA, the authors further proposed an approximation algorithm of FFA that can be done efficiently. The empirical results demonstrated the unique properties of FFA compared to other SOTA algorithms.

**Weaknesses:**

The work can potentially be improved from either the theoretical side or the experimental side.

From theory direction:
It is not clear about the time complexity of the approximating formal feature attribution and the approximation error compared to the actual FFA.

From the experimental direction:
The authors highlighted the advantages of FFA compared to existing algorithms. However, the evidence showing this is the case in this work is limited.
1. The authors mentioned the "unsoundness of explanations" exists in prior work. It would be great to specify what type of unsoundness the authors specifically refer to, and explain why the FFA can increase such "soundness". The same justification is needed for the "out-of-distribution sampling" performance compared to existing works
2. All the experiments on the comparison between FFA and other SOTA XAI algorithms are based on real-life examples. It is hard to evaluate, if two methods disagree with each other, which one is better.  Why don't the authors start with clean synthetic data, where the ground truth is known, and show where FFA generates better results than other XAI algorithms? And further show that the proposed approximation FFA algorithm did a better job than other XAI algorithms.

**Questions:**

1. The authors mentioned that "Formal explanation also tend to be larger than their model-agnostic counterparts...", can you formally define what "larger" means in this context?
2. The authors mentioned, "We argue that formal feature attribution is hard for the second level of the polynomial hierarchy". Can you clarify what " the second level of the polynomial hierarchy" specifically means here? When you say "hard", what do you mean specifically? Can you quantify the hardness or show any examples?

---

### Official Review · Reviewer_vzsC · 2023-10-30

**Soundness:** 3 good
**Presentation:** 2 fair
**Contribution:** 2 fair
**Rating:** 5
**Confidence:** 3

**Summary:**

This paper proposes formal feature attributions (FFA). The proposed method assigns attribution scores to a feature corresponding to the share of valid logical explanations in which a feature appears. An algorithm to compute approximate and exact FFA is presented.

**Strengths:**

*  Definitions and somewhat intuitive and are clear to follow.
*  The algorithm to compute FFA seems to work and scales up to tree models and datasets of MNIST size. The experiments highlight that is is feasible to compute good approximations of FFA within a realistic runtime budget

**Weaknesses:**

*  Lack of related work. As there is no dedicated related work section, the relation to prior work can be strengthened. As this paper is dealing with feature attributions, there is a multitude of related approaches that could potentially be mentioned [1, 2, 3, 4]. There should be much more space invested to motivate why FFA are relevant and what practival problems they could potentially solve.
*   The propositions don’t have proofs. I cannot follow the argumentation leading up to Proposition 1 and 2. While they may be intuitive to experts, I think the exposition is insufficient for the average reader to follow. I agree that conference papers have limited space, but the proofs certainly need to be amenable to verification also by non-experts. I don’t think is insufficient to make imprecise statements like “this is a direct consequence of a result in Huang et al.”. The exact result should be named, and restated in the appendix together with an explanation for how the proposition can be derived from it. This also helps the reader to assess the novelty of this claims.
*  Scope: The scope of the method is limited to Boosted-Trees. While I agree that this is a relevant model for tabular data, it is not the only one. This may limit the paper’s potential impact.
*   The evaluation metrics use FFA’s as a ground truth (Table 2) and seem to hypothesize that LIME and SHAP should be approximating FFAs. The disagreement is obvious as SHAP and lime are meant to serve different purposes and are derived from different desiderata
*  Some claims are exaggerated. For instance, “In this paper we define the first formal approach to feature attribution” (Conclusion) is too strong and incorrect in its current form. Other attribution methods such as Shapley Values or IG [2] also have formal grounding, and can therefore also be considered formal approaches to feature attribution. I think such overly strong statements should be avoided.

**Minor:**
Clarity of the write-up: Unfortunately, there are many writing/grammar issues in the paper, particularly in the introduction. I encourage the authors to check the writeup again. For instance, already the first sentence of the introduction seems a bit puzzling, as it does not mention what is growing and appears to be overcomplicated. What is meant by “We argue that formal feature attribution is hard for the second level of the polynomial hierarchy.” (Section 1)? Usually, a comma is employed after “e.g.” and “i.e.”.

## Summary

A work on formal explainability that proposes a novel feature attribution method grounded in formal explanations (FFA). The algorithm to compute FFAs seems interesting and shows that these attributions can be computed for problems up to MNIST size. Unfortunately, there are many competing notions of attribution, the technical novelty seems limited, and the motivation for this approach is not extensive. Furthermore, following the draft is complicated by some missing explanations for the propositions and minor writing issues. While I don’t think the work has substantial flaws, I still lean towards reject as of now.


--------------------------------
**References**

[1] Lapuschkin, S., Binder, A., Montavon, G., Klauschen, F., Müller, K.-R., and Samek, W. On pixel-wise explanations for non-linear classiﬁer decisions by layer-wise relevance propagation. PloS one, 10(7):e0130140, 2015.

[2] Sundararajan, M., Taly, A., and Yan, Q. Axiomatic attribution for deep networks. In International Conference on Machine Learning, pp. 3319–3328. PMLR, 2017.

[3] Kasneci, G. and Gottron, T. Licon: A linear weighting scheme for the contribution of input variables in deep artiﬁcial neural networks. In Proceedings of the 25th ACM International on Conference on Information and Knowledge Management, pp. 45–54, 2016.

[4] Petsiuk, V., Das, A., and Saenko, K. Rise: Randomized input sampling for explanation of black-box models. In Proceedings of the British Machine Vision Conference (BMVC), 2018.

**Questions:**

*   What is the purpose of the weighted attributions, if they are – as claimed in the beginning of the experimental section – almost identical?
*   Figure 5 and 6 (particularly) show that FFA are not necessarily meaningful for humans to understand the prediction. They suffer from the same drawbacks of all feature attribution explanations that reason on a pixel level, which is not a level human users usually reason in. Can the authors give a concrete use-case where FFA can however unfold advantages over other feature attribution techniques?

---

### Official Review · Reviewer_LjFe · 2023-10-31

**Soundness:** 3 good
**Presentation:** 3 good
**Contribution:** 3 good
**Rating:** 6
**Confidence:** 3

**Summary:**

The authors argue that despite the success of AI and ML, there are still critical issues that need to be addressed, such as model brittleness, fairness, and interpretability. They propose a new approach to feature attribution based on formal explanation enumeration, which aims to provide a more sound and reliable method for understanding the weight of a feature in making a classification decision.

The authors begin by discussing the two major lines of work in XAI: feature selection methods and feature attribution techniques. They note that while these approaches show promise, they are susceptible to a range of critical issues, including explanation unsoundness and out-of-distribution sampling. They argue that a formal approach to feature attribution is needed to address these limitations.

The authors then introduce their proposed approach, which is based on the proportion of abductive explanations in which a feature occurs to weigh its importance. They show that this approach can be used to compute feature attribution for many classification problems, and when exact computation is not possible, effective approximations can be used. They compare their approach to existing heuristic approaches to feature attribution and show that they do not always agree with their proposed method.

**Strengths:**

1. The authors provide a theoretical framework for understanding the soundness and reliability of their approach. They show that their approach can be used to compute feature attribution exactly for many classification problems, which is a significant contribution. However, they also acknowledge that exact computation may not always be possible, and in such cases, effective approximations can be used. It would be useful for the authors to provide more details on the approximations used and their accuracy.

2. The authors compare their approach to existing heuristic approaches to feature attribution and show that they do not always agree with their proposed method. This is an important contribution, as it highlights the limitations of existing methods and the need for a more formal approach to feature attribution. However, it would be useful for the authors to provide more details on the specific heuristic approaches they compared their method to and how they conducted the comparison.

**Weaknesses:**

1. While the authors propose a new approach to feature attribution based on formal explanation enumeration, it is not entirely clear how this approach differs from existing methods. The authors briefly mention that their approach is based on the proportion of abductive explanations in which a feature occurs to weigh its importance, but it is not clear how this differs from existing methods such as LIME and SHAP.

2. The authors focus primarily on the theoretical aspects of their approach and do not provide sufficient empirical evaluation. While the authors show that their approach can be used to compute feature attribution exactly for many classification problems, they do not provide sufficient empirical evaluation to demonstrate the practical usefulness of their approach.

3. The authors do not provide sufficient details on the approximations used when exact computation is not possible. While the authors mention that effective approximations can be used, they do not provide sufficient details on the specific approximations used and their accuracy.

**Questions:**

1. The authors propose a new approach to feature attribution based on formal explanation enumeration. Can you provide more details on how this approach differs from existing methods such as LIME and SHAP? What are the specific advantages of your approach?

2. The authors show that their approach can be used to compute feature attribution exactly for many classification problems. Can you provide more details on the types of classification problems for which exact computation is possible? What are the limitations of your approach in terms of the types of problems it can handle?

3. The authors mention that effective approximations can be used when exact computation is not possible. Can you provide more details on the specific approximations used and their accuracy? How were these approximations evaluated?

---

### Official Review · Reviewer_oi1W · 2023-11-04

**Soundness:** 3 good
**Presentation:** 4 excellent
**Contribution:** 3 good
**Rating:** 5
**Confidence:** 2

**Summary:**

The paper builds on formal explainability, introducing Formal feature attribution. In formal explainability, the explanations are equated with abductive explanations (AXp's), which are subset-minimal sets of features formally proved to suffice to explain an ML prediction given a formal representation of the classifier of interest; AXp's requires a formal reasoner to check all combinations of feature values, which makes AXp’s long and there can be various AXp’s for a single data instance. The paper introduces formal feature attribution of a given feature as the proportion of abductive explanations where it occurs i.e it is the percentage of (formal abductive) explanations that make use of a particular feature $i$. To compute FFA AXp enumeration is required, the paper posed an approximation to that: instead of using MARCO targeting AXp’s the paper proposes the target contrastive explanations (CXp) enumeration with AXp’s as dual explanations the exact algorithm is shown in Algorithm 1.

Experiments:

- The paper tests FFA for gradient boosted trees for image and tabular datasets and compares there results with LIME, TreeSHAP, and KernalSHAP.

- For dataset the paper considered MNIST but only as a binary classification problem, PneumoniaMNIST and 11 tabular datasets the maximum size of the input features for tabular dataset is 15.

- The paper compared the exact formal attribution with the baselines on tabular dataset and showed that none of the baselines agree with FFA.

- The paper then compared approximate FFA constrained by time with exact FFA and showed that they can gets good FFA approximations even if they only collect AXp’s for a short time.

**Strengths:**

### Originality -- Strong:
The idea of calculating feature attribution on top of formal explanations is original and novel.


### Clarity -- Strong:
- The paper is well-written and easy to follow.
- The proposed method is well-motived and related work is clearly explained.

**Weaknesses:**

Significance -- Weak:

- The main drawback of the method is that it depends on the ability to compute Axp's. Although the paper does propose an approximation (algorithm 1) it is still can be quite expensive in practice.

- Most current AI methods use deep neural networks, all experiments done in the paper used boosted tree this is because FFA requires the ML to have some first-order logic, which is not practical so the use of the proposed method in practice will be very limited.

- The results and time comparison between exact FFA, approximate FFA and baselines is done on datasets with very small input space and binary classification problem. These results will probably not generalize for larger input space for example,  if LIME takes < 1 second for 10x10 images scaling for larger images like 256×256 lime will still be pretty quick if but FFA_10 will probably not give any useful results.

**Questions:**

- Why were the datasets reduced to a binary classification problem rather than a multiclass problem?

- For tabular data, can FFA be used on larger datasets trained like Rossmann Store Sales [1] i.e much larger input space if so how does it compare to LIME and others in terms of computation?

- For the wide use of FFA, practitioners need to be able to use FFA on black-box neural networks, how can FFA (or a variation of the currently proposed FF) be used to explain neural networks like TabNet [2] with no first-order logic ?


[1] https://www.kaggle.com/c/rossmann-store-sales/data

[2] TabNet: Attentive Interpretable Tabular Learning